# Factors Associated with Tooth Loss in General Population of Bialystok, Poland

**DOI:** 10.3390/ijerph19042369

**Published:** 2022-02-18

**Authors:** Katarzyna Gabiec, Joanna Bagińska, Wojciech Łaguna, Ewa Rodakowska, Inga Kamińska, Zofia Stachurska, Marlena Dubatówka, Marcin Kondraciuk, Karol Adam Kamiński

**Affiliations:** 1Private Dental Clinic ‘Lux-Dent’ Stomatologia, 15-668 Bialystok, Poland; gabieck@o2.pl; 2Department of Dentistry Propaedeutics, Medical University of Bialystok, 15-295 Białystok, Poland; 3Faculty of Computer Science, Bialystok University of Technology, 15-351 Białystok, Poland; wojciech.laguna@gmail.com; 4Department of Clinical Dentistry-Cariology Section, University of Bergen, 5020 Bergen, Norway; ewa.rodakowska@uib.no; 5Department of Integrated Dentistry, Medical University of Bialystok, 15-276 Białystok, Poland; inga.kaminska@umb.edu.pl; 6Department of Population Medicine and Lifestyle Diseases Prevention, Medical University of Białystok, 15-269 Białystok, Poland; zofia.stachurska@umb.edu.pl (Z.S.); marlena.dubatowka@umb.edu.pl (M.D.); marcin.kondraciuk@umb.edu.pl (M.K.); karol.kaminski@umb.edu.pl (K.A.K.)

**Keywords:** tooth loss, risk factors, age, body mass index, fasting blood glucose level, smoking, cross-sectional study

## Abstract

Background: The aim of this study was to assess risk factors for tooth loss in the population of the city of Bialystok, in north-eastern Poland, taking into account the entire population and different age groups. The study included 1138 subjects divided into three subgroups: 20–44 years, 45–64 years, and 65–79 years. Participants were classified according to the number of teeth lost (0–8 vs. 9–28). Socio-economic variables, smoking history, and dental habits were collected through a questionnaire. Medical examinations provided data on the body mass index and the fasting blood glucose level. Data were statistically analysed using Mann-Whitney U, Student’s t, chi^2^ tests, and binary logistic regression, *p* < 0.05. Results: For the general population, being female (OR 1.38, 1.07–1.79, *p* = 0.015), having secondary education (OR 4.18, Cl 2.97–5.87, *p* < 0.000), higher body mass index (OR 1.13, Cl 1.10–1.17, *p* < 0.000), higher fasting blood glucose level (OR 1.03 1.03–1.04, *p* < 0.000), being former smoker (OR 1.72, Cl 1.29–2.31, *p* < 0.000), ever smoker (OR 1.69, Cl 1.29–2.20, *p* < 0.000), current smoker (OR 1.62, Cl 1.15–2.29, *p* < 0.006), longer smoking period (OR 1.11, Cl 1.09–1.14, *p* < 0.000), last visit to the dentist over a year ago (OR 1.92, Cl 0.44–2.58, *p* < 0.000) and tooth brushing less than two times a day (OR 1.6, Cl 1.14–2.23, *p* < 0.006) were associated with losing more than 8 teeth. In the subgroup aged 20–44 years, only smoking duration was a risk factor for tooth loss (*p* = 0.02). For the middle-aged and oldest groups, education level (respectively *p* < 0.001, and *p* = 0.001), body mass index (respectively, *p* < 0.001, and *p* = 0.037), smoking status ever/former/current (respectively *p* < 0.001 and *p* = 0.002), smoking status never/ever (respectively *p* < 0.001 and *p* = 0.009), smoking duration (*p* < 0.001) were related to tooth loss. Additionally, in the elderly group, fasting blood glucose level (*p* = 0.044) and frequency of dental visits (*p* = 0.007) were related to tooth loss. We concluded that in the evaluated population, tooth loss was associated with socio-demographic, medical, and behavioural factors.

## 1. Introduction

According to the World Health Organization (WHO), a key aspect of oral health is the lifelong maintenance of functional dentition, understood as dentition consisting of no less than 20 teeth, without the need for tooth replacement [1,2]. Tooth loss results in functional, aesthetic, and social impairments, may decrease an individual’s quality of life and could be an effective determinant of population oral health [3,4]. The World Dental Federation (FDI) and WHO established Global Oral Health Goals for 2020, stipulating among other targets that the proportion of 35–44 and 65–74 year olds with functional dentition should increase in each population [5].

The prevalence of tooth loss has constantly decreased during the last decades, especially in developed countries [3,6], but disparities among countries and regions are may still be observed. The main reasons for missing teeth are dental caries, periodontal diseases, trauma, and orthodontic extractions [7,8,9]. Untreated dental caries is considered the main cause of tooth loss except for adults older than 80 years, and another leading reason is periodontitis [7,8,9,10,11]. Some reports indicate that causes of tooth extraction differ according to age and sex [12], and the distribution of missing teeth in dental arches depends on the cause. Several factors are associated with tooth loss: age, gender, socio-behavioural factors, oral health behaviours, availability, and quality of dental service [4]. There is still no clear evidence as to whether oral conditions or socio-behavioural factors should be considered as the most significant risk factors [6]. The association between tooth loss and non-communicable diseases (NCDs) was investigated, and its relation to cardiovascular disease, stroke, diabetes, metabolic syndrome, dementia, depression, and all-cause mortality was confirmed [13,14,15,16,17,18]. NCDs share common risk factors as dental conditions leading to dental extraction [19,20]. Poor general health (e.g., mental and physical disability, depression) and aging may also contribute, as a predisposing factor, to the neglect of oral hygiene and may increase the risk of severe caries and periodontal disease [2,21,22,23,24].

Previous studies on risk factors for tooth loss showed that the importance of factors related to the number of missing teeth varied according to the population assessed [4,25,26,27]. The majority of these studies, also conducted on the Polish population, concerned the elderly [28,29,30,31]. Learning about risk factors in younger populations may be important in reducing tooth loss over a lifetime. Therefore, the study aimed to assess risk factors for tooth loss among the population of the city of Bialystok, in north-eastern Poland, taking into account the entire population and different age groups.

## 2. Materials and Methods

### 2.1. Study Population

Based on the Mayor’s Office database, a random sampling was conducted to determine a study population. The data on age and gender distribution in the entire population are available from the Polish Statistical Office upon request [32]. The study was approved by the Ethics Committee of the University of Bialystok, Poland (R-i-002/108/2016) in conformity with the Declaration of Helsinki. The sample size was 3806, and was stratified according to age and gender distribution across the entire population (Figure 1). The inclusion criteria were ages between 20 and 79 and being a citizen of Bialystok, Poland.

Between July 2017 and May 2021, 3246 subjects from the drawn group were invited to participate in the study. For the present study, the minimum sample size was determined using an online tool available at https://www.naukowiec.org/dobor.html (accessed on 17 August 2021) with a 383 person level. The calculation was based on the following criteria: in 2017, the average number of residents of Bialystok aged 20–79 years was 207,676, due to lack of epidemiological data on the prevalence of tooth loss in the entire population, the percentage of people with at least one missing tooth (the fraction size) was established at 50%, standard values for the confidence level −95% and the maximum error −5% were adopted [33].

### 2.2. Data Acquisition

The survey questionnaire was self-administered, printed, and prepared in Polish. It consisted of a broad range of closed and open questions concerning socio-economic situation, medical history, family history, health habits, and quality of life. For the present study, the following closed questions were included: age, gender, educational level, subject’s smoking history: being never/ever-smoker (NS/ES), never/former/current-smoker (NS/FS/CS), duration of smoking and their dental habits: frequency of tooth brushing and dental visits were collected using a questionnaire survey. Medical examinations provided data on body mass index (BMI, kg/m^2^). The fasting blood glucose level (FBG, mg/dL) measurement was a part of the general blood test. Medical and dental examinations were performed on the same day.

Dental examinations were conducted by four calibrated dentists under conditions of an epidemiological inquiry (with the use of artificial light and without the use of a saliva ejector or air jet). A full mouth oral examination was performed to estimate the number of lost teeth. Teeth missing due to dental caries, periodontal, and trauma reasons were counted as lost. Third molars, unerupted teeth, and teeth extracted due to orthodontic reasons were excluded from the analysis.

### 2.3. Statistical Analysis

Descriptive statistics for quantitative variables were presented as mean and standard deviations and as counts and frequencies for qualitative variables. Comparisons of variables between subgroups were conducted using the U Mann-Whitney test or t-Student test for quantitative variables and the chi^2^ test for qualitative variables. Statistical hypotheses were verified at a 0.05 significance level. A binary logistic regression analysis was performed to estimate the odds ratio (OR) for the association between risk factors and having more than 8 teeth lost. The Statistica 13.3. (StatSoft Polska Sp. z o.o., Cracow, Poland) was used for all calculations.

## 3. Results

Out of those invited to the survey, 1196 (36.85%) responded. A total of 58 individuals were excluded, either because they had refused to participate in the dental portion or there was another reason for the lack of data on missing teeth. As a result, 1138 individuals were included in the study group; the final response rate was 35.05%. The study group consisted of 503 (44.2%) men and 635 (55.8%) women, with a mean age of 48.8 ± 15.38 years. Participants were divided into three subgroups based on their age: aged 20 to 44 years (young group), 45 to 64 years (middle-aged group), and 65 years and older (elderly group). The number of participants in each group was respectively: 501, 409, and 228. For the analysis, participants were dichotomized into two groups according to the number of teeth lost due to caries, periodontal disease, and dental trauma (0–8 vs. 9–28).

Table 1 and Table 2 present the assessed risk factors for tooth loss for the whole population and subgroups by age. Age, gender, education level, BMI, FBG, cigarette smoking, duration of smoking, frequency of tooth brushing, and dental visits were factors related to tooth loss in the general population. In the 20–44 years subgroup, only duration of smoking was a risk factor for tooth loss. In the middle-aged group, variables favouring the preservation of more own teeth were lower BMI, no smoking history, and smoking for a shorter period. In the oldest group, factors associated with the number of lost teeth were higher BMI, higher FBG level, being ES and CS, long-term smoking, and irregular dental visits.

In all groups, regardless of age and number of lost teeth, the mean BMI was above 25. The mean BMI in middle-aged and older participants who lost more than 8 teeth was 29.1 and was significantly higher compared to those who lost less than 8 teeth (27.3, *p* < 0.001 in the middle-aged subgroup and 27.6, *p* = 0.003 in the older subgroup, respectively).

In relation to the general population, subjects with a greater number of lost teeth had higher mean FBG levels (above normal range) compared to the other group, 109.4 ± 25.7 vs. 98.3 ± 16.4 (*p* < 0.001). In the youngest subjects, the mean FBG level was within normal limits (below 99 mg/dl) irrespective of the number of lost teeth. In other age subgroups, the mean parameter was above normal. However, only in the oldest subgroup, an association between mean FBG levels and the number of lost teeth was statistically confirmed (*p* = 0.044).

Smoking history was associated with a higher number of lost teeth in the general population (*p* < 0.001). ES accounted for 70.2% of those with severe tooth loss in the middle-aged subgroup (*p* < 0.001) and 62.6% in the oldest group (*p* = 0.009). Furthermore, in both groups, an association between being NS, FS, and CS and the chance of preserving dentition was observed.

Another risk factor for tooth loss was the duration of smoking. In the general population, this period was significantly shorter in subjects who lost less than 8 teeth compared to the other group (12.4 ± 8.9 vs. 27.5 ± 14, *p* < 0.001). Moreover, it was the only variable associated with the level of tooth loss in all age groups assessed (*p* = 0.02 for participants aged 20–44, and *p* < 0.001 for the other two subgroups).

Most subjects brushed their teeth at least twice a day (83.6%). Almost half of the participants (48.4%) declared their last dental visit within the last 6 months. Regarding these variables, only in the elderly group was there a statistical association between tooth loss and the frequency of dental visits. Only 25% of the participants in this age group who had more than twenty teeth preserved declared irregular visits to a dentist.

Table 3 presents an association between risk factors and having more than 8 teeth lost for the general population. It was shown that a lower education level had the most significant impact on the number of lost teeth (OR 4.18, Cl 2.97–5.87, *p* < 0.000), followed by dental visits less than once a year (OR 1.92, Cl 0.44–2.58, *p* < 0.000), having smoking habit: being a former smoker (OR 1.72, Cl 1.29–2.31, *p* < 0.000), ever smoker (OR 1.69, Cl 1.29–2.20, *p* < 0.000), current smoker (OR 1.62, Cl 1.15–2.29, *p* < 0.006) and brushing teeth less than two times a day (OR 1.6, Cl 1.14–2.23, *p* < 0.006). Moreover, for every unit of BMI, the odds of being in the 9–28 loss group increased by 1.13 times (Cl 1.10–1.17, *p* < 0.000), and for every additional unit of FBG, the odds of being in the group with tooth loss between 9 and 28 was 1.03 higher (1.03–1.04, *p* < 0.000). Each year of smoking increased the odds of losing more than 8 teeth by 1.11. (Cl 1.09–1.14, *p* < 0.000).

## 4. Discussion

Demographic and socio-economic factors were evaluated as risk factors of tooth loss in several studies [3,25,26,27,28,29,30,31,34]. The association between tooth loss and participants’ age found in this study is consistent with previous reports [3,35]. A negative impact of caries and periodontal disease accumulating throughout life results in an increasing number of missing teeth with age, with the peak incidence in the seventh decade of life [3]. Carmen et al. [35] found that the average number of missing teeth increased by 5% for each year of the patient’s age. In this study, the evaluation of risk factors was conducted in three separate age groups: 20–44, 45–64, and 65–79 years. In the evaluated sample, the gender contributed as a tooth loss risk factor for the whole population (OR 1.38, Cl 1.07–1.97), but not for particular age groups, similarly to the Mexican population [35]. Kassebaum et al. [3] found that globally gender differences in tooth loss decreased between 1990 and 2010. In our study, having higher education reduced the risk of severe tooth loss by 41.8 times. A university level education contributed as a protective factor in the middle-aged and elderly groups, in accordance with de Miguel-Infante et al. [36] results that a university degree is associated with a lower risk of periodontitis. The Study of Health in Pomerania (SHIP) found that low education and low income were associated with tooth loss [27,37]. Kim and Kim [38], in their study on self-rated poor oral health conducted for the same age groups as in the present study, found that the educational level was a more important covariate among young and middle-aged people (20–44 and 45–64), and income played a greater role among adults aged 65 and over.

Many studies confirmed that obesity correlated with fewer teeth [39,40], however, there are reports of no such link [41]. In the study by Őstberg et al. [40], the association between tooth loss and general and abdominal obesity in people under 60 years was independent of age and gender, socio-economic factors, lifestyle, and co-morbidities. The relationship between oral health and obesity is multifactorial. Undoubtedly, high consumption of added sugars may lead to increased BMI and severe dental caries [42,43,44,45]. It was found that middle-aged Thai adults with frequent consumption of sugary snacks were more prone to tooth loss [46]. Increased consumption of sugar-sweetened beverages, a marker of fermentable sugars intake, was observed in young and elderly people with tooth loss [9,47]. Obesity and associated diseases were found to be risk factors for periodontitis [48,49]. The pathways of links between obesity and periodontitis remain unclear, with a likely bidirectional influence [49]. Meisel et al. [39] suggested sex-specific differences in the incidence of periodontitis and tooth loss in obese subjects related to different CRP levels. In the present group, higher BMI was observed in those who lost more than 8 teeth in the general population and in both the middle-aged and oldest age groups. We found that the chance of losing more than 8 teeth increased with every additional BMI unit by 1.13. The present study indicates that not only obesity but also BMI at the upper limit of overweight are variables associated with the risk of tooth loss. From a clinical perspective, however, there were no great differences in mean BMI, as it ranged between 25 and 29.9, indicating that Bialystok residents, in general, tended to be overweight.

Hyperglycaemia was observed in the middle-aged and the oldest groups. In elderly subjects, it was statistically proved that the prevalence of severe tooth loss was higher in those with higher levels of fasting glucose blood levels. A direct association between the number of missing teeth and diabetes was confirmed [9,16,50]. Liljestrand et al. [16] found missing more than 9 teeth associated with an incident of cardiovascular disease, diabetes, and death of any cause. A similar conclusion can be drawn from the National Health and Nutrition Examination Survey, 2003–2004, conducted in the USA population aged 50 and more. The mean number of missing teeth in people with diagnosed diabetes was 9.8 compared to 6.8 in those without the disease [51]. Moreover, according to that study, twenty percent of cases of edentulism in the USA were linked to diabetes. Greenblatt et al. [50] found that uncontrolled diabetes significantly increased the likelihood of missing >9 teeth, including edentulousness, especially in younger individuals (18–44 years). A possible connection between tooth loss and diabetes is via periodontal disease [52]. Data from a South Korean Nationwide Health Screening Program showed that the FBG correlated with periodontitis and tooth loss but not with dental caries [53]. In the study of de Miguel-Infante et al. [36], the subjects with diabetes had an increased ratio of periodontal disease by 22%. Diabetes mellitus is also one of the co-morbidities associated with obesity; links between tooth loss and increased BMI have been explained above.

The present study confirmed that cigarette smoking was an important risk factor regarding tooth loss. Habitual smokers made up the majority in the middle-age and oldest age groups with tooth loss of more than 8 teeth. Moreover, the number of years of smoking was strongly associated with a greater number of lost teeth in all age groups. The data from the literature are inconsistent [26,28,54]. Some studies align with our results [28,54], but other studies did not confirm the relationship between smoking status and tooth loss [26]. Similä et al. [54] confirmed that both intensity and duration of smoking contributed as risk factors of tooth loss. The modern oral health approach states that oral conditions share common risk factors with other non-communicable diseases, and tobacco smoking is one of such variables [20]. The correlation between tobacco use and an increased rate of dental caries was previously reported. However, the mechanism of promoting caries by nicotine products is still unclear [55]. Periodontal disease is more common, has a more severe course, and therefore presents a higher risk of treatment failure in smokers [56]. In addition, this relationship is dose-dependent [57]. There are several mechanisms of negative effects of smoking on periodontal tissues, and one of them is the alteration of alveolar bone metabolism. Increased alveolar bone resorption due to osteoclast formation and activation results in an increased tooth loss [57,58], and smoking cessation may improve periodontal health [56]. We found that with longer duration of smoking increased the chance of being in the group with 9–28 teeth lost. Surprisingly, our study revealed that being FS was associated with a higher risk of losing more than 8 teeth (higher than being CS (OR 1.72, Cl 1.29–2.31 and OR 1.62, Cl 1.15–2.29, respectively). This may be explained by the fact that we did not assess the number of pack-years, an indicator of the accurate exposure to smoking.

Our observation that people over 64 years of age brushed their teeth less frequently than young and middle-aged people and that they were prone to miss regular dental visits is consistent with Canadian data for the same age stratification [59]. In the present study, the prevalence of tooth loss was more often in participants who brushed teeth less than two times a day and were irregular dental attenders. However, with respect to the age subgroup, only the frequency of dental visits was significant for the oldest group. According to Tiwari et al. [26], regular visits to a dentist impacted tooth retention in older adults. However, in the Mexican population, the frequency of dental attendance did not influence the number of missing teeth, but brushing teeth less than every day was associated with it [35].

The strengths of this study include a random selection of participants and a study population that is three times the minimum sample size. An additional strength of the present study is the inclusion of adults from all life stages. Most studies on risk factors for tooth loss focused on middle-aged or elderly individuals only. The low response rate (35.05%) should be considered as a limitation, especially for a cross-sectional survey, as there is a possibility that non-respondents differed from respondents. Another limitation of cross-sectional studies constitutes the difficulties in distinguishing between association and causation in the exposure-outcome relationship. For data collected through the questionnaire, there may be a bias in responses regarding health habits and a ‘fatigue effect’ [60] due to the length of the survey tool. Another limitation is that only people living in the city were surveyed, so the data obtained cannot be directly transferred to the entire population living in north-eastern Poland.

## 5. Conclusions

In the study population, tooth loss was associated with socio-demographic, medical, and behavioural factors that varied depending on the age group. The duration of smoking was the only factor associated with tooth loss in all groups. Smoking cessation counselling should be an integral part of dental treatment and prevention, especially in young patients.

## Figures and Tables

**Figure 1 ijerph-19-02369-f001:**
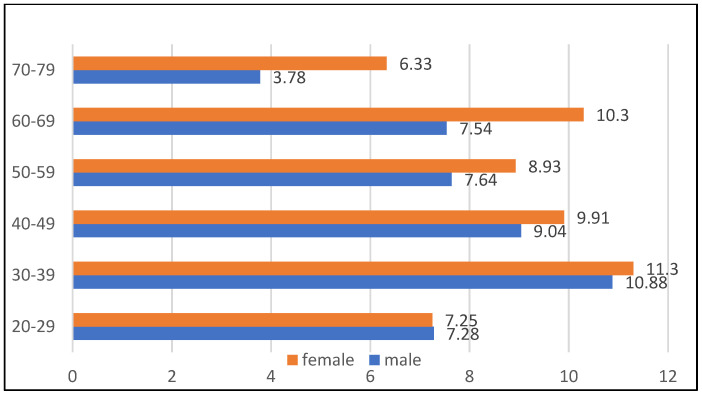
The distribution of the population drawn for the survey by age and sex.

**Table 1 ijerph-19-02369-t001:** The tooth loss (0–8 vs. 9–28) and mean age, BMI, FBG, and duration of smoking. (*p* < 0.05, * U Mann–Whitney test, ** t–Student test) BMI—body mass index, FBG—fasting blood glucose level.

Risk Factor		Overall		0–8 Teeth Lost		9–28 Teeth Lost		*p*
		*N*	Mean (SD)	*N*	Mean (SD)	*N*	Mean (SD)	
Age	General population	1138	48.8 (15.4)	804	42.59 (13.0)	334	63.9 (8.8)	<0.001 *
Body mass index (kg/m^2^)	General population	1138	26.9 (4.0)	804	26.0 (4.6)	334	29.1 (5.1)	<0.001 *
	20–44	501	25.2 (4.5)	492	25.2 (4.5)	9	27.2 (5.2)	0.207 *
	45–64	409	27.0 (4.9)	258	27.3 (4.6)	151	29.1 (5.2)	<0.001 *
	>64	228	28.7 (4.8)	54	27.6 (4.2)	174	29.1 (4.9)	0.037 **
Fasting glucose blood level (mg/dL)	General population	1134	101.6 (20.2)	800	98.3 (16.4)	334	109.4 (25.7)	<0.001 *
	20–44	498	94.9 (15.5)	489	94.9 (15.6)	9	95.3 (8.1)	0.75 *
	45–64	409	104.6 (18.9)	258	103.5 (16.6)	151	106.5 (22.2)	0.288 *
	>64	227	110.9 (26.1)	53	104.8 (13.9)	174	112.7 (28.5)	0.044 *
Duration of smoking (years)	General population	586	17.5 (13.0)	389	12.4 (8.9)	197	27.5 (14.0)	<0.001 *
	20–44	257	10.2 (6.9)	252	10.1 (6.8)	5	17.2 (4.1)	0.02 *
	45–64	208	21.4 (12.5)	114	16.9 (10.6)	94	26.8 (12.6)	<0.001 *
	>64	121	26.2 (15.4)	23	15.6 (10.1)	98	28.6 (15.5)	<0.001 *

**Table 2 ijerph-19-02369-t002:** Prevalence of tooth loss according to demographic characteristics and smoking and oral habits. (*p* < 0.05, chi^2^ test) NS—never-smoker, FS—former smoker, CS—current smoker, ES—ever smoker.

Risk Factor		Overall *N* (%)	0–8 Teeth Lost *N* (%)	9–28 Teeth Lost *N* (%)	*p*-Value
Gender	General population	*n* = 1138	*n* = 804	*n* = 334	
	male	503 (44.2)	374 (46.5)	129 (38.6)	0.015
	female	635 (55.8)	430 (53.5)	205 (61.4)	
	20–44	*n* = 501	*n* = 492	*n* = 9	
	male	238 (47.5)	233 (47.4)	5 (55.6)	0.626
	female	263 (52.5)	259 (52.6)	4 (44.4)	
	45–64	*n* = 409	*n* = 258	*n* = 151	
	male	169 (41.3)	114 (44.2)	55 (36.4)	0.626
	female	240 (58.7)	144 (55.8)	96 (63.6)	
	>64	*n* = 228	*n* = 54	*n* = 174	
	male	96 (42.1)	27 (50.0)	69 (39.7)	0.179
	female	132 (57.9)	27 (50.0)	105 (60.3)	
Education level	General population	*n* = 801	*n* = 602	*n* = 199	
	secondary	333 (41.6)	199 (33.1)	134 (67.3)	<0.001
	university	468 (58.4)	403 (66.9)	65 (32.7)	
	20–44	*n* = 384	*n* = 376	*n* = 8	
	secondary	124 (32.3)	120 (31.9)	4 (50.0)	0.279
	university	260 (67.7)	256 (68.1)	4 (50.0)	
	45–64	*n* = 277	*n* = 190	*n* = 87	
	secondary	123 (44.4)	65 (34.2)	58 (66.7)	<0.001
	university	154 (55.6)	125 (65.8)	29 (33.3)	
	>64	*n* = 140	*n* = 36	*n* = 104	
	secondary	86 (61.4)	14 (38.9)	72 (69.2)	0.001
	university	54 (38.6)	22 (61.1)	32 (30.8)	
Smoking status (NS/FS/CS)	General population	*n* = 1138	*n* = 804	*n* = 334	
	Never smoker	489 (43.0)	375 (46.6)	114 (34.1)	0.001
	Former smoker	416 (36.5)	273 (34.0)	143 (42.8)	
	Current smoker	233 (20.5)	156 (19.4)	77 (23.1)	
	20–44	*n* = 501	*n* = 492	*n* = 9	
	Never smoker	215 (42.9)	211 (42.9)	4 (44.5)	0.695
	Former smoker	168 (33.5)	166 (33.7)	2 (22.2)	
	Current smoker	118 (23.6)	115 (23.4)	3 (33.3)	
	45–64	*n* = 409	*n* = 258	*n* = 151	
	Never smoker	178 (43.5)	133 (51.5)	45 (29.8)	<0.001
	Former smoker	144 (35.2)	84 (32.6)	60 (39.7)	
	Current smoker	87 (21.3)	41 (15.9)	46 (30.5)	
	>64	*n* = 228	*n* = 54	*n* = 174	
	Never smoker	96 (42.1)	31 (57.4)	65 (37.3)	0.002
	Former smoker	104 (45.6)	23 (42.6)	81 (46.6)	
	Current smoker	28 (12.3)	0 (0.0)	28 (16.1)	
Smoking status (NS/ES)	General population	*n* = 1138	*n* = 804	*n* = 334	
	Never smoker	489 (43.0)	375 (46.6)	114 (34.1)	<0.001
	Ever smoker	649 (57.0)	429 (53.4)	220 (65.9)	
	20–44	*n* = 501	*n* = 492	*n* = 9	
	Never smoker	215 (42.9)	211 (42.9)	4 (44.4)	0.925
	Ever smoker	286 (57.1)	281 (57.1)	5 (55.6)	
	45–64	*n* = 409	*n* = 258	*n* = 151	
	Never smoker	178 (43.5)	133 (51.6)	45 (29.8)	<0.001
	Ever smoker	231 (56.5)	125 (48.4)	106 (70.2)	
	>64	*n* = 228	*n* = 54	*n* = 174	
	Never smoker	96 (42.1)	31 (57.4)	65 (37.4)	0.009
	Ever smoker	132 (57.9)	23 (42.6)	109 (62.6)	
Frequency of tooth brushing	General population	*n* = 1096	*n* = 781	*n* = 315	
	less than two times per day	180 (16.4)	113 (14.5)	67 (21.3)	0.006
	two times per day or more	916 (83.6)	668 (85.5)	248 (78.7)	
	20–44	*n* = 489	*n* = 480	*n* = 9	
	less than two times per day	70 (14.3)	69 (14.4)	1 (11.1)	0.782
	two times per day or more	419 (85.7)	411 (85.6)	8 (88.9)	
	45–64	*n* = 391	*n* = 250	*n* = 141	
	less than two times per day	57 (14.6)	34 (13.6)	23 (16.3)	0.466
	two times per day or more	334 (85.4)	216 (86.4)	118 (83.7)	
	>64	*n* = 216	*n* = 51	*n* = 165	
	less than two times per day	53 (24.5)	10 (19.6)	43 (26.1)	0.349
	two times per day or more	163 (75.5)	41 (80.4)	122 (73.9)	
Time of last dental visit	General population	*n* = 1113	*n* = 790	*n* = 323	
	within last 6 months	539 (48.4)	404 (51.1)	135 (41.8)	<0.001
	between last 6 and 12 month	234 (21.0)	179 (22.7)	55 (17.0)	
	longer than 12 months	340 (30.6)	207 (26.2)	133 (41.2)	
	20–44	*n* = 492	*n* = 483	*n* = 9	
	within last 6 months	247 (50.2)	241 (49.9)	6 (66.7)	0.5
	between last 6 and 12 month	110 (22.4)	108 (22.4)	2 (22.2)	
	longer than 12 months	135 (27.4)	134 (27.7)	1 (11.1)	
	45–64	*n* = 402	*n* = 255	*n* = 147	
	within last 6 months	203 (50.5)	134 (52.6)	69 (46.9)	0.099
	between last 6 and 12 month	90 (22.4)	61 (23.9)	29 (19.7)	
	longer than 12 months	109 (27.1)	60 (23.5)	49 (33.4)	
	>64	*n* = 219	*n* = 52	*n* = 167	
	within last 6 months	89 (40.6)	29 (55.8)	60 (35.9)	0.007
	between last 6 and 12 month	34 (15.5)	10 (19.2)	24 (14.4)	
	longer than 12 months	96 (43.9)	13 (25.0)	83 (49.7)	

**Table 3 ijerph-19-02369-t003:** Association between risk factors and having more than 8 teeth lost (binary logistic regression, OR—odds ratio, Cl—confidence level).

Risk Factor		No. (%) Participants	OR (95% Cl)	*p*
Gender	male	129 (38.62)	1	
	female	205 (61.38)	1.38 (1.07–1.79)	0.015
Education level	university	65 (32.66)	1	
	secondary	134 (47.34)	4.18 (2.97–5.87)	0.000
Smoking status (NS/FS/CS)	NS	114 (34.13)	1	
	FS	143 (42.81)	1.72 (1.29–2.31)	0.000
	CS	77 (23.05)	1.62 (1.15–2.29)	0.006
Smoking status (NS/ES)	NS	114 (34.13)	1	
	ES	220 (65.87)	1.69 (1.29–2.20)	0.000
Frequency of tooth brushing	two times per day or more	248 (78.73)	1	
	less than two times per day	67 (21.27)	1.6 (1.14–2.23)	0.006
Time of last dental visit	within last 6 months	135 (41.8)	1	
	between last 6 and 12 month	55 (17.02)	0.92 (0.64–1.32)	0.067
	longer than 12 months	133 (41.18)	1.92 (1.44–2.58)	0.000
Body mass index		334 (100)	1.13 (1.10–1.17)	0.000
Fasting glucose blood level		334 (100)	1.03 (1.03–1.04)	0.000
Duration of smoking		197 (100)	1.11 (1.09–1.14)	0.000

## Data Availability

Data available on request from the authors.

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
