# Peer review of "Factors Associated with Tooth Loss in General Population of Bialystok, Poland"

_ijerph, 2022, doi:10.3390/ijerph19042369_

Round 1

Reviewer 1 Report

The present paper reports the results of a study conducted in the city of Bialystok, Poland, in which the authors analysed the relationship between several socioeconomic or behaviour factors and tooth loss in subject of different ages. 

It would be useful to perform a minor check of English. 

Line 18: abbreviations (BMI and FBG) should be defined also in the abstract 

Line 19: "chi2" should be "chi2"

Line 28: it would be useful to indicate other key words regarding the evaluated factors or the nature of the study, in order to ease the identification of the paper

Line 36: WHO abbreviation was already defined in line 31, so only the abbreviation WHO should remain at this point

Lines 56-63: this section can be merged with the introduction section

Lines 71-86: information regarding the number of subjects that responded to the invitation, that were excluded form the analysis (and the reasons for exclusion), that were finally included (and their characteristics: age, gender), and the number of subjects in each group should be reported in the results section.

Line 94: it would be useful to give more details regarding the misuration of MBI and FBG: for example, were they performed by the same investigator? Were they performed in conjunction with the dental examination?

Line 94: " by four calibrated dentists in dental surgeries" did you mean "by four calibrated dentists"? Please clarify

Line 104: "chi2" should be "chi2"

Line 108: "on assessed evaluated risk factors" should be "on the assessed risk factors"

Line 157: 48.8% of subjects declared their last dental visit within the last 6 months, but they do not represent the majority of the included subject as stated.

Author Response

We would like to thank the Reviewer for the careful analysis of our paper and thoughtful comments. 

The present paper reports the results of a study conducted in the city of Bialystok, Poland, in which the authors analysed the relationship between several socioeconomic or behaviour factors and tooth loss in subject of different ages. 

It would be useful to perform a minor check of English. – The manuscript has been proofread by a professional English translator.

Line 18: abbreviations (BMI and FBG) should be defined also in the abstract  - The abbreviation have been defined in the abstract.

Line 19: "chi2" should be "chi2"  - The correct notation has been used.

Line 28: it would be useful to indicate other key words regarding the evaluated factors or the nature of the study, in order to ease the identification of the paper. -  We have added additional key words

Line 36: WHO abbreviation was already defined in line 31, so only the abbreviation WHO should remain at this point. - We have corrected the notation.

Lines 56-63: this section can be merged with the introduction section. -  We have merged both sections.

Lines 71-86: information regarding the number of subjects that responded to the invitation, that were excluded form the analysis (and the reasons for exclusion), that were finally included (and their characteristics: age, gender), and the number of subjects in each group should be reported in the results section.  - Such information has been transferred to the Results section.

Line 94: it would be useful to give more details regarding the misuration of MBI and FBG: for example, were they performed by the same investigator? –  Were they performed in conjunction with the dental examination?  BMI calculation was included in the medical examination and FGB measurement was performed as part of the general blood test. Medical and dental examinations were performed on the same day by the authors of this paper. -  Such information has been included in the paper.

Line 94: " by four calibrated dentists in dental surgeries" did you mean "by four calibrated dentists"? Please clarify  - It has been corrected.

Line 104: "chi2" should be "chi2" - The correct notation has been used.

Line 108: "on assessed evaluated risk factors" should be "on the assessed risk factors"  -  It has been corrected.

Line 157: 48.8% of subjects declared their last dental visit within the last 6 months, but they do not represent the majority of the included subject as stated.  - It has been corrected.

Reviewer 2 Report

Factors associated with tooth loss in general population of Bialystok, Poland

Risk factors for tooth loss were investigated in 1138 subjects from different age groups (20-44, 45-64 and 65-79 years). Age, gender, education level, BMI, FBG, history of smoking, duration of smoking, frequency of tooth brushing, and dental visits appeared to be factors associated with tooth loss.

Although the topic of the research is not original, it is interesting that the results of a large group of participants with different ages are presented. The manuscript is clearly written and easy to follow. The results confirm findings of previous studies. The data, data analysis, and presentation are appropriate. It may be better not to use abbreviations without explanation in the abstract (BMI, FBG). I have no further comments.

Author Response

We would like to thank the Reviewer for the careful analysis of our paper and thoughtful comments. 

Although the topic of the research is not original, it is interesting that the results of a large group of participants with different ages are presented. The manuscript is clearly written and easy to follow. The results confirm findings of previous studies. The data, data analysis, and presentation are appropriate. It may be better not to use abbreviations without explanation in the abstract (BMI, FBG). I have no further comments. - The use of abrreviations has been corrected.

Reviewer 3 Report

Thank you very much for submitting this manuscript.

In general readers may be confused by "functional dentition" (tooth number >20) and "tooth loss" (Table 1: 0-8 and 9-28 tooth loss). Authors are advised to use one of them throughout the article.

Since "Teeth extracted due to orth reasons were excluded from the analysis" so the relationship between "functional dentition" and "tooth loss 0-8 / 9-28", the relationship of "full dentition=functional dentiton + tooth loss" may not valid.

Another issue for this manuscript is the reason of not using regression model to investigate the effect of multi-variables on tooth loss.

Abstract: Should mention full name of BMI and FBG

Materials and methods:

1. Authors may describe more detail of the "stratified randomization" e.g. by random number tables etc.

2. Authors are suggested to include a reference of the age and GENDER distribution in the entire population

3. More information about the questionnaire may be included (e.g. language and the format of questions such as open end question or multiple choices)

4. The last few sentence of first paragraph of M&M describe the rationale of sample size in this study. More information is needed such as i) the type of statistical tests such as t test/chi square test/ Mann-Whitney U test etc. ii) the rationale of using percentage of people with at least one tooth missing 50% and maximum error at 5% etc.

5. Please cite a study that has same age classification as this study if possible.

6. For the reference 1, it also classify the functional dentition as type I/II/III and IV. Do authors refer to any type(s) in this study?

7. With over 1000 sample size, it is easy to achieve statistical significance. Table 1: BMI 45-64 age group, the differences in BMI are small (27.0/27.3/29.1). Are this BMI clinical significant? Moreover for Mann-Whitney tests, authors are advised to show if there is differences between two or three groups. In this example, I guess 29.1>27.3=27.0?

Table 1, authors may add the number of subjects (N) for all age groups in the column "overall".

Authors are advised to run a regression analysis to see which factor(s) are confounding.

Author Response

We would like to thank the Reviewer for all valuable comments on the manuscript.

Thank you very much for submitting this manuscript.

In general readers may be confused by "functional dentition" (tooth number >20) and "tooth loss" (Table 1: 0-8 and 9-28 tooth loss). Authors are advised to use one of them throughout the article. - It has been corrected.

Since "Teeth extracted due to orth reasons were excluded from the analysis" so the relationship between "functional dentition" and "tooth loss 0-8 / 9-28", the relationship of "full dentition=functional dentiton + tooth loss" may not valid.  - As suggested, the term 'functional dentition' has been excluded. For the purposes of this paper, individuals who had lost any tooth for orthodontic reasons were considered to be without tooth loss. 

 Another issue for this manuscript is the reason of not using regression model to investigate the effect of multi-variables on tooth loss. – The binary logistic regression analysis has been performed (Table 3).

Abstract: Should mention full name of BMI and FBG . - It has been corrected.

Materials and methods:

  1. Authors may describe more detail of the "stratified randomization" e.g. by random number tables etc. – More details of the sampling method have been added to the methodology. The stratification of the sample has been shown in Fig. 1.
  2. Authors are suggested to include a reference of the age and GENDER distribution in the entire population - Data are available at the website of the Polish Statistical Office (reference included) 
  3. More information about the questionnaire may be included (e.g. language and the format of questions such as open end question or multiple choices). - The description of the questionnaire has been expanded.
  4. The last few sentence of first paragraph of M&M describe the rationale of sample size in this study. More information is needed such as i) the type of statistical tests such as t test/chi square test/ Mann-Whitney U test etc. ii) the rationale of using percentage of people with at least one tooth missing 50% and maximum error at 5% etc. - The rationale of our assumption has been explained. Due to unknown prevalence of tooth loss in the entire population, the percentage of people with at least one missing tooth (the fraction size) was established at 50%, and standard values for the confidence level – 95% and the maximum error - 5% were adopted. We have included a reference to that.
  5. Please cite a study that has same age classification as this study if possible. - We have added a references with the same age distribution from population studies on oral health in Canada and South Korea (38, 50, 59).
  6. For the reference 1, it also classify the functional dentition as type I/II/III and IV. Do authors refer to any type(s) in this study? - We did not refer to these types of functional dentition. The term “functional dentition” has been removed as suggested.
  7. With over 1000 sample size, it is easy to achieve statistical significance. Table 1: BMI 45-64 age group, the differences in BMI are small (27.0/27.3/29.1). Are this BMI clinical significant? Moreover for Mann-Whitney tests, authors are advised to show if there is differences between two or three groups. In this example, I guess 29.1>27.3=27.0?- - In the Discussion, a sentence has been added to explain that from a clinical point of view the differences in mean BMI between the groups were small. A clarification has been added to the title of Table 1 to explain that the Mann-Whitney U test was used to compare the two groups (0-8 lost teeth vs. 9-28 lost teeth).

Reviewer 4 Report

In the abstract section, please do not use the short forms.

If you can perform the regression analysis for identifying the significant variables in the order of its association with tooth loss, it would be better.

Rest of the sections are fine.

Author Response

We would like to thank the Reviewer for all valuable comments on the manuscript.

In the abstract section, please do not use the short forms. - The abbreviation have been defined in the abstract.

If you can perform the regression analysis for identifying the significant variables in the order of its association with tooth loss, it would be better. - A binary logistic regression analysis has been performed (Table 3).

Rest of the sections are fine.

Round 2

Reviewer 3 Report

Thanks Author for revise the manuscript

Abstract:

  1. Line 20: (BMI), should remove bracket
  2. binary logistic regression analysis should be included in the method and result
  3. Result should present with p-values
  4. For binary logistic regression, authors are suggested to list the binary grouping in the abstract. e.g. (University/secondary)
  5. Result should present the positive/negative association with tooth loss. E.g. Longer smoking duration associated with tooth loss etc.
  6. “The number of associated variables was related to the age.” I wonder if this is confounding factor after performing the regression.

Keywords:

  1. “BMI”, “fasting blood glucose level” some abbreviation some full name?

Introduction:

  1. FDI is “World Dental Federation” or “World dental Association“?
  2. “There is still no clear evidence as to whether oral conditions or socio-behavioural factors should be considered as the most significant risk factors” and “Learning about risk factors in younger populations may be important in reducing tooth loss over a lifetime. Therefore, the aim of this study was to assess risk factors for tooth loss in different age groups in the population of Bialystok city, north-eastern Poland”. Is this the main aim of this manuscript? If yes,
  3. https://bdl.stat.gov.pl/bdl/dane/teryt/tablica This link is not work in my browser.
  4. “unknown prevalence of tooth loss in the entire population the percentage of people with at least one missing tooth (the fraction size) was established at 50%”. Do authors mean that it is established that 50% of entire population the percentage of people with at least one missing tooth?

Result: In general the results and tables are difficult to understand

  1. Line 144 to line 155. The level of BMI and FBG should also present in table 1. Line 165. Duration of smoking in years were not presented in any of the tables. We only know the duration from 7 to 13 years (Line 169 to 170)
  2. Line 161: “retained more than 20 teeth” were not simply equal to “0 to 8 teeth loss” because of unerupted teeth and teeth extracted due to orthodontic reasons.

Table 1

  1. Regarding the single row of “age”, “Fasting blood glucose level”, “BMI” and “Duration of smoking” what are the groups compared? Did authors compare “mean” or “proportions” or “co-relation”? Mann-Whitney tests and student t tests are comparing means.
  2. What does the legend “association” means?

Table 3

  1. There are two “smoking status”. Smoking duration (years) (significant in abstract line 22) was not observed.
  2. P value should be presented

Discussion should address the weakness/limitations of this study includes the cross sectional nature of this study, the accuracy of questionnaire and the absence of data such as number of pack of cigarettes per day etc. have not been discussed.

Author Response

First of all, we would like to thank the Reviewer for an insightful analysis of our manuscript and the valuable comments.

Abstract:

  1. Line 20: (BMI), should remove bracket.
  2. binary logistic regression analysis should be included in the method and result
  3. Result should present with p-values
  4. For binary logistic regression, authors are suggested to list the binary grouping in the abstract. e.g. (University/secondary)
  5. Result should present the positive/negative association with tooth loss. E.g. Longer smoking duration associated with tooth loss etc.
  6. “The number of associated variables was related to the age.” I wonder if this is confounding factor after performing the regression.

Response: The Abstract has been rewritten.

Keywords: “BMI”, “fasting blood glucose level” some abbreviation some full name?

Response: Key word has been changed.

Introduction:

  1. FDI is “World Dental Federation” or “World dental Association“?

Response: Wrong name has been corrected.

  1. “There is still no clear evidence as to whether oral conditions or socio-behavioural factors should be considered as the most significant risk factors” and “Learning about risk factors in younger populations may be important in reducing tooth loss over a lifetime. Therefore, the aim of this study was to assess risk factors for tooth loss in different age groups in the population of Bialystok city, north-eastern Poland”. Is this the main aim of this manuscript? If yes,

Response: The aim of the study has been written.

  1. https://bdl.stat.gov.pl/bdl/dane/teryt/tablica This link is not work in my browser.

Response:

Indeed there may be problems in directly accessing data on the distribution of the population of the city of Bialystok by sex and age. To gain access, one has to go through several search steps or request the data from this institution. The article notes that these data are available upon request from [email protected]

  1.  “unknown prevalence of tooth loss in the entire population the percentage of people with at least one missing tooth (the fraction size) was established at 50%”. Do authors mean that it is established that 50% of entire population the percentage of people with at least one missing tooth?

Response:

As mentioned in this sentence - there are no epidemiological data on the prevalence of tooth loss in the entire population of Bialystok. In such cases, the assumption that the fraction size is 50% gives the largest minimum study population and is the recommended treatment. Incorrect reference numbering has been changed in this sentence (33 instead of 32).

Result: In general the results and tables are difficult to understand

Response:

We simplified the Results paragraph by removing some of the information shown in Tables 1 and 2.

  1. Line 144 to line 155. The level of BMI and FBG should also present in table 1. Line 165. Duration of smoking in years were not presented in any of the tables. We only know the duration from 7 to 13 years (Line 169 to 170)

Response:

The mean levels of BMI and FBG as well as duration of smoking are presented in Table 1. The layout of the table has been changed to make it better readable. The inaccurate sentence regarding differences in mean duration of smoking between both groups has been removed.

  1. Line 161: “retained more than 20 teeth” were not simply equal to “0 to 8 teeth loss” because of unerupted teeth and teeth extracted due to orthodontic reasons.

Response: This sentence has been removed.

 Table 1

  1. Regarding the single row of “age”, “Fasting blood glucose level”, “BMI” and “Duration of smoking” what are the groups compared? Did authors compare “mean” or “proportions” or “co-relation”? Mann-Whitney tests and student t tests are comparing means.

Response

We compared subjects with tooth loss 0-8 vs subjects with tooth loss 9-28. The data presented in Table 1 are average values. This is specified in the table. We have added an explanation in the title of Table 1.

The layout of all tables has been changed to make them better readable.

  1. What does the legend “association” means?

Response

The title of this table has been changed.

Table 3

  1. There are two “smoking status”. Smoking duration (years) (significant in abstract line 22) was not observed.

Response

The results of logistic regression for BMI, FGB and duration of smoking have been added to Table 3.

  1. P value should be presented

Response

P values have been added to Table 3.

Discussion should address the weakness/limitations of this study includes the cross sectional nature of this study, the accuracy of questionnaire and the absence of data such as number of pack of cigarettes per day etc. have not been discussed.

Response:

The paragraph on limitations has been rewritten. The lack of pack-years as an indicator of accurate smoking exposure was highlighted in the Discussion section relating to smoking.